# Balanced Cellular and Humoral Immune Responses Targeting Multiple Antigens in Adults Receiving a Quadrivalent Inactivated Influenza Vaccine

**DOI:** 10.3390/vaccines9050426

**Published:** 2021-04-23

**Authors:** Esther Dawen Yu, Alba Grifoni, Aaron Sutherland, Hannah Voic, Eric Wang, April Frazier, Natalia Jimenez-Truque, Sandra Yoder, Sabrina Welsh, Stacey Wooden, Wayne Koff, Buddy Creech, Alessandro Sette, Ricardo da Silva Antunes

**Affiliations:** 1Center for Infectious Disease and Vaccine Research, La Jolla Institute for Immunology, La Jolla, CA 92037, USA; dyu@lji.org (E.D.Y.); agrifoni@lji.org (A.G.); asutherland@lji.org (A.S.); hannah.voic12@ncf.edu (H.V.); ericwang@lji.org (E.W.); afrazier@lji.org (A.F.); 2Vanderbilt Vaccine Research Program, Vanderbilt University Medical Center, Nashville, TN 37232, USA; natalia.jimenez@vumc.org (N.J.-T.); sandra.yoder@vumc.org (S.Y.); buddy.creech@vumc.org (B.C.); 3Human Vaccines Project, New York, NY 10119, USA; SWelsh@humanvaccinesproject.org (S.W.); wkoff@humanvaccinesproject.org (W.K.); 4Merck & Co., Inc., Kenilworth, NJ 07033, USA; swooden@globalhealthalliance.org; 5Department of Medicine, Division of Infectious Diseases and Global Public Health, University of California, San Diego, La Jolla, CA 92037, USA

**Keywords:** T cells, protein immunodominance, cytokine polarization, influenza viruses, vaccine

## Abstract

The role of T cell immunity has been acknowledged in recent vaccine development and evaluation. We tested the humoral and cellular immune responses to Flucelvax^®^, a quadrivalent inactivated seasonal influenza vaccine containing two influenza A (H1N1 Singapore/GP1908/2015 IVR-180 and H3N2 North Carolina/04/2016) and two influenza B (Iowa/06/2017 and Singapore/INFTT-16-0610/2016) virus strains, using peripheral blood mononuclear cells stimulated by pools of peptides overlapping all the individual influenza viral protein components. Baseline reactivity was detected against all four strains both at the level of CD4 and CD8 responses and targeting different proteins. CD4 T cell reactivity was mostly directed to HA/NA proteins in influenza B strains, and NP/M1/M2/NS1/NEP proteins in the case of the Influenza A strains. CD8 responses to both influenza A and B viruses preferentially targeted the more conserved core viral proteins. Following vaccination, both CD4 and CD8 responses against the various influenza antigens were increased in day 15 to day 91 post vaccination period, and maintained a Th1 polarized profile. Importantly, no vaccine interference was detected, with the increased responses balanced across all four included viral strains for both CD4 and CD8 T cells, and targeting HA and multiple additional viral antigens.

## 1. Introduction

The influenza virus is an enveloped virus with a relatively simple structure. A total of three influenza proteins are incorporated in the viral membrane: the hemagglutinin (HA), neuraminidase (NA), and M2 proton channel, while the virion core contains the matrix (M1), nuclear export (NEP), non-structural (NS1) proteins, and eight viral ribonucleoproteins (vRNP). Each vRNP contains the viral polymerases with PA, PB1, and PB2 subunits, the nucleoprotein (NP) and a single strand of negative-sense viral RNA (vRNA) [1]. Owing to the genomic instability, influenza viruses gain frequent antigenic alterations through antigen drift (point mutations in the viral genome) and antigen shift (gene segment exchange between viral genomes) [2]. This is the reason why a yearly vaccine is required. Most licensed influenza vaccines are based on inactivated trivalent (TIV) or quadrivalent (QIV) designs that contain purified HA proteins of two influenza A strains and one or two influenza B strains. The combination of strains is standardized annually in compliance with World Health Organization (WHO) [3] and Committee for Medicinal Products for Human Use (CHMP) recommendations (EU) [4], or United States Public Health Service requirements (in the USA) [5], to provide protection against the strains expected to circulate in the upcoming influenza season.

Flucelvax Quadrivalent^®^ (Seqirus, Holly Springs, NC, USA) is an inactivated influenza vaccine that is produced in a mammalian cell line (Madin Darby canine kidney (MDCK) cells) [6]. For the 2018–2019 influenza season, Flucelvax was designed to contain two influenza A (H1N1 Singapore/GP1908/2015 IVR-180 and H3N2 North Carolina/04/2016) and two influenza B virus strains (Iowa/06/2017 and Singapore/INFTT-16-0610/2016) [5]. Compared to conventional egg-based influenza vaccines, its production can be expanded to large-scale within a short timeframe without egg-specific adaptions and mutations [7]. Since licensure in 2016, studies have shown that Flucelvax is potentially superior to egg-based vaccines in preventing influenza infection both in terms of real-life effectiveness, as well as in the protective efficacy demonstrated during clinical trials [8].

Different from recombinant influenza vaccines that contain pure HA antigens, Flucelvax contains 15 microgram (mcg) hemagglutinin (HA) from each of four strains with additional viral proteins (NA, M1, M2, NP, NEP, NS1, PA, PB1, and PB2) not more than 240 mcg per 0.5 mL dose [5]. Current vaccines are licensed based on World Health Organization-approved centers that test the vaccine efficacy solely based on standard measures of humoral immunity. Therefore, most studies, including Flucelvax, have focused on evaluating humoral [9], as well as HA-specific CD4 dependent neutralizing antibody responses [10]. The antibody responses to any of the viral proteins other than HA are not considered or measured because their role is traditionally dismissed from an immunological and vaccine efficacy perspective. The exception is the viral NA, which is attracting growing interest because NA-specific antibodies are considered an independent correlate of protection [11].

Even though extensive studies have reported that both CD4 and CD8 T cells recognize conserved influenza epitopes from viral proteins other than HA [12,13], the study of the cellular immune responses elicited by influenza core protein components in vaccination settings, and particularly in response to Flucelvax immunization are lacking. Hence, studying how non-HA viral proteins contained in Flucelvax contribute to the vaccination response is of interest. Accordingly, we used a functional assay that allows simultaneous detection of influenza-specific CD4 and CD8 T cell responses to protein components of various influenza A and B strains in peripheral blood mononuclear cells (PBMC), using pools of peptides overlapping the individual viral protein components. The current study performs a comprehensive analysis of (i) the pre-existing immune reactivity and (ii) the vaccine-specific cellular and humoral responses in adults receiving inactivated mammalian cell-based influenza vaccine, Flucelvax.

## 2. Materials and Methods

### 2.1. Longitudinal Study Design

The overall purpose of this longitudinal study is to investigate the cellular and humoral antigen specific immune responses in adults receiving inactivated mammalian cell-based influenza vaccine for the current baseline influenza Kinetics of the Immune Response to Inactivated Influenza Vaccine in Healthy Adults (KIRV) study without adjuvant. This cohort is part of a broader study from the Human Vaccine Project (HVP) [14] and the Universal Influenza Vaccine Initiative, which is also studying the tissue-specific immune responses that vaccination elicits. Accordingly, a total of 10 healthy males and non-pregnant females aged 18 to 49 years old were enrolled through the Vanderbilt Vaccine Research Program at Vanderbilt University Medical Center (VUMC) during March–October, 2019. The study was approved by the VUMC Human Subjects Protection Program and was registered at clinicaltrials.gov (NCT03743688). Each participant provided informed consent and was assigned a study identification number with clinical information recorded. Subjects with acute illness, cancer, known HIV, hepatitis B or C infection, any other contraindicative medical condition, recent immunoglobulin or other blood products transfusion, history of using immunosuppressive or immunomodulating therapy including chemotherapy, radiation therapy and corticosteroids, history of allergy or other severe reaction following previous immunization, received or plan to receive live or inactivated influenza vaccine or any other experimental or interventional agent within 120 days before or after vaccination were excluded.

Each participant was administered one dose of the inactivated mammalian cell-based Flucelvax Quadrivalent influenza vaccine intramuscularly, and 60 mL blood draws were obtained for immunological characterization during each visit. For each subject, 3 blood draws were obtained. One pre-immunization (D1, at baseline before injection) sample, one 14 days post-vaccination (D15) sample and one 90 days post-vaccination (D91) sample. In all cases, PBMCs were isolated from whole blood by density gradient centrifugation according to manufacturer instructions (Ficoll-Hypaque, Amersham Biosciences, Uppsala, Sweden) and cryopreserved for further analysis.

### 2.2. Design of Antigens Utilized in the Evaluation of Cellular Immunity

During the 2018–2019 flu season, the Flucelvax Quadrivalent vaccine was formulated to contain a total of 60 microgram (mcg) hemagglutinin (HA) (with other viral proteins ≤240 mcg) per 0.5 mL dose in the recommended ratio of 15 mcg HA of each of the following four influenza strains [5]: the two influenza A virus strains are the A/Singapore/GP1908/2015 IVR-180 (H1N1) (an A/Michigan/45/2015-like virus), and the A/North Carolina/04/2016 (H3N2) (an A/Singapore/INFIMH-16-0019/2016-like virus). The two influenza B virus strains are the B/Iowa/06/2017 (a B/Colorado/06/2017-like virus) and the B/Singapore/INFTT-16-0610/2016 (a B/Phuket/3073/2013-like virus).

Previous studies developed and validated the used of large peptide pools to evaluate cellular immune responses, denominated MegaPools (MP) [15]. Epitope Flu MPs were performed based on sequential lyophilization of overlapping 15-mers by 10 pooled by protein combinations as previously reported [16]. Protein sequences for the corresponding vaccine strains have been extracted from the NIAID Influenza Research Database (IRD) [17] or GISAID [18]. For each strain, four different MPs were generated combining the following proteins: HA/NA, NP/M1/M2/NS1/NEP, PA/PB1385, and PB1376/PB2). More specifically, for each strain we designed a HA/NA MP (204–207 peptides per each strain), a PA-PB1 MP (216–218 peptides per each strain) a PB1-PB2 MP (213–227 peptides per each strain) and a MP addressing all other proteins (NP/M1/M2/NS1/NEP 232–256 peptides per each strain). Individual peptides were synthesized by A&A (San Diego, CA, USA) and pooled by protein combinations and resuspended to a final concentration of 1 mg/mL in DMSO. Detailed information of the MPs composition and peptide sequences were listed in Appendix A. We also utilized, as additional controls, two previously described MPs encompassing T cell epitopes from Bordetella pertussis (PT) [19,20] and Cytomegalovirus (CMV) [15].

### 2.3. Functional T Cell Assay

To evaluate cellular immune responses, PBMC from each subject at the three time points were thawed, rested in culture overnight, and stimulated with the flu MPs described above at the final concentration of 1 µg/mL for 6 h in in 96-wells U bottom plates at 2 × 10^6^ PBMCs per well. An equimolar volume of DMSO was used as negative control. Golgi-Plug containing brefeldin A (BD Biosciences, San Diego, CA, USA) was added after 3 h into the culture. At the end of the 6 h of stimulation, intracellular staining was performed, as previously described [21]. Cells were stained with surface markers for 30 min at 4 ℃ followed by fixation with 4% paraformaldehyde (Sigma–Aldrich, St. Louis, MO, USA) at 4 ℃ for 10 min. Intracellular staining was incubated at room temperature for 30 min after cells permeabilization with saponin. The antibody panel utilized in the ICS staining is shown in Appendix A. Data were acquired in a BD LSRII Flow Cytometer (BD Biosciences, San Jose, CA, USA) using BD FACSDiva software version 6.1.3 (BD Biosciences, San Jose, CA, USA) and analyzed by FlowJo X Software (version 10) (Tree Star, Ashland, OR, USA).

The gating strategy utilized was shown in Appendix A, the gates applied for the identification of IFNγ, Granzyme B, IL-4, or IL-10 production on the total population of CD4+ or CD8+ T cells were defined according to the cells cultured with DMSO for each individual. Antigen-specific responses were defined as effector producing cells that express the CD154 activation marker. Responses from a representative donor are shown on the right side of the Appendix A.

### 2.4. Detection of Antibodies to Seasonal Influenza Viruses and Vaccine

Serological testing for detection of antibodies to four seasonal influenza viruses strains included in the Flucelvax Quadrivalent vaccine (A/Michigan/45/2015-like virus, A/Singapore/INFIMH-16-0019/2016-like virus, B/Colorado/06/2017-like virus, and B/Phuket/3073/2013-like virus) were performed at Infectious Diseases Research Facility–Frederick (Southern Research, Frederick, MD, USA). The antibody titers were measured using both Microneutralization assay (MN) and Hemagglutination inhibition assay (HAI) following standard protocols.

### 2.5. Statistical Analysis

Experimental data were analyzed by GraphPad Prism Version 8 (GraphPad Software, San Diego, CA, USA) and Microsoft Excel Version 16.16.27 (Microsoft, Redmond, WA, USA). The statistical details of the experiments are provided in the respective figure legends. Normality of distribution was accessed by Shapiro–Wilks test. Parametric data from the longitudinal study were analyzed by paired Student’s *t* test (two-tailed) to compare with baseline, one-way ANOVA to compare between groups corrected with Tukey’s test to adjust for multiple comparisons, and Pearson correlation analysis. Non-parametric data were analyzed by Wilcoxon test (two-tailed) to compare with baseline, Kruskal–Wallis test adjusted with Dunn’s test for multiple comparisons to compare between groups, and Spearman correlation analysis. Data were plotted as median with interquartile range for non-parametric data and mean with SEM for parametric data. *p* values < 0.05 (after adjustment if indicated) were considered statistically significant.

### 2.6. Study Approval

This study was approved by the Human Subjects Protection Program of Vanderbilt University Medical Center, Nashville, Tennessee (IRB Protocol no. 181619). Each participant provided informed consent and was assigned a study identification number with clinical information recorded.

## 3. Results

### 3.1. Vaccine Study Design

A total of 10 healthy males and non-pregnant females aged 18 to 49 years old were enrolled at Vanderbilt University Medical Center (VUMC) during March to October 2019. This cohort was enrolled to participate in the Universal Influenza Vaccine Initiative (see Material and Methods). The demographics of these subjects are listed in Table 1. In total, four males and six females were recruited. Mean age at the time of immunization was 35.60 ± 3.54 years. Mean BMI was 26.04 ± 1.24. The recruited cohort roughly matched the racial distribution in Tennessee with 90% Caucasian and 10% African American [22]. Each participant received one dose of the inactivated mammalian cell-based Flucelvax Quadrivalent influenza vaccine intramuscularly, and 3 blood specimens, drawn on D1 (pre-immunization), D15 (14 days post-vaccination), and D91 (90 days post-vaccination) were used to specifically assay for T cell responses. No seasonal flu vaccination was administered 4 months prior to sample collection. The overall demographic features define this cohort suitable for longitudinal analysis of the immune responses induced by Flucelvax.

### 3.2. Strategy to Measure Cellular Responses

In 2018–2019 flu season, the Flucelvax Quadrivalent vaccine incorporates inactivated viruses derived from two influenza A (H1N1 Singapore/GP1908/2015 IVR-180 and H3N2 North Carolina/04/2016) and two influenza B virus strains (Iowa/06/2017 and Singapore/INFTT-16-0610/2016). In addition to HA, Flucelvax contains additional viral proteins (NA, M1, M2, NP, NEP, NS1, PA, PB1, and PB2). To comprehensively monitor T cell reactivity against the vaccine components, for each strain, four different pools of peptides (MegaPools; MP) were designed based on the sequences of the various influenza proteins.

More specifically, for each strain we designed a HA/NA MP (204–207 peptides per each strain), a PA-PB1 MP (216–218 peptides per each strain), a PB1-PB2 MP (213–227 peptides per each strain), and a MP spanning all other proteins (NP/M1/M2/NS1/NEP) (232–256 peptides per each strain) (Appendix A). The specific MPs were prepared by sequential lyophilization of overlapping 15-mers by 10 pooled by protein combinations, as previously reported [16]. We also utilized two previously described MPs as additional controls encompassing T cell epitopes from *Bordetella pertussis* (PT) [19,20] and cytomegalovirus (CMV) [15].

To evaluate cellular responses, PBMC obtained from each subject at the three time points described above were stimulated with various MPs also described above, and the number of responsive CD4 and CD8 T cells were measured by staining of cell surface phenotype markers and intracellular cytokine and cytotoxic markers (IFNγ, Granzyme B, IL-4 and IL-10). Gating strategy from one representative donor is shown in Appendix A. As expected, a strong response was observed in the case of the PMA + Io positive control, as compared to the DMSO only negative control. In the case of the stimulation with the HA/NA flu MP, IFNγ polarized response was observed, which indicated a Th1 response.

### 3.3. Overall CD4 T Cell Responses to the Flucelvax Quadrivalent Vaccine

Total CD4 T cell reactivity at any time point was defined as the total number of cells that produce any of the 4 effector responses (IFNγ, Granzyme B, IL-4 and IL-10) to the different MPs from all 4 influenza strains. Overall CD4 T cell responses are summarized in Figure 1A, and the results expressed as numbers of responding CD4 cells per million of total CD4 cells (left panel) or as fold-change compared to the baseline (D1) (right panel).

The mean overall influenza-specific CD4 T cell reactivity at baseline was 1414 ± 287 (numbers of responding CD4 cells per million of total CD4 cells). Increased reactivity on D15 was observed in 8 out of the 10 (80%) subjects, with increased mean reactivity of 1966 ± 338 (*p* = 0.036) and a fold-change of 1.5 ± 0.2 (*p* = 0.021). The responses were still elevated in eight (80%) subjects on D91 (*p* = 0.029), with mean reactivity of 1801 ± 233 and a fold-change of 1.5 ± 0.2. The individual trends of changes in CD4 T cells responses were shown in Appendix A, and it is interesting to note that one person did not have an elevated CD4 T cell reactivity at D15 or D91 timepoints. These results demonstrate that Flucelvax vaccination increased influenza-specific CD4 T cell reactivity in most subjects.

### 3.4. Overall CD8 T Cell Responses to the Flucelvax Quadrivalent Vaccine

Similarly, the overall baseline influenza-specific CD8 T cell reactivity was 301 ± 63 (numbers of responding CD8 cells per million of total CD8 cells) (Figure 1B). Overall, 9 out of 10 (90%) subjects had vaccine responses on D15 with mean response reactivity of 517 ± 112 (*p* = 0.048) and fold change of 2.2 ± 0.3 (*p* = 0.008). Elevated responses on D91 were still observed among 6 out the 10 (60%) subjects, with further increased mean reactivity of 546 ± 141 and a fold-change of 2.4 ± 0.8, but the differences were no longer significant from baseline. The individual trends of changes in CD8 T cells responses were shown in Appendix A. These results show that Flucelvax immunization increased CD8 T cell reactivity, which is remarkable as subunit vaccines are generally not considered to be effective in inducing CD8 responses [23].

### 3.5. Overall Cellular Responses to PT and CMV Control Antigens

We also analyzed CD4 T cell reactivity to two previously described MPs encompassing T cell epitopes from Bordetella pertussis (PT) [19,20] and cytomegalovirus (CMV) [15] as controls. The results are shown in Appendix A. For the CMV MP, 6 participants were seronegative, and did not show any reactivity to the CMV MP, as expected (data not shown). Among the 4 seropositive subjects, median CD4 T cell reactivity at time zero was 958 (range: 718 to 4104, as numbers of responding CD4 cells per million of total CD4 cells), which was also unchanged post vaccination, as expected (Appendix A. In the case of PT MP, significant CD4 T cell reactivity (stimulation index (SI) >2) was observed among 8 out of the 10 donors. Median CD4 T cell reactivity at time zero was 20.5 (range: 0.0- to 55, as numbers of responding CD4 cells per million of total CD4 cells), which, as expected, did not significantly change following vaccination (Appendix A. CD8 responses to CMV and PT control conditions were not assessed since the MP selection was exclusively based on prediction of HLA class II promiscuous binding peptides for CMV, which is also known to elicit very little CD8 T cell reactivity to PT [24]. Overall, these results show that pre-existing PT and CMV CD4 T cell reactivity were not modulated by Flucelvax vaccination.

### 3.6. Serological Responses to Vaccination

Serological responses to the vaccine were also evaluated using the microneutralization (MN) assay, which is based on the ability of serum antibodies to prevent influenza virus infection of mammalian cells in vitro [25]. The results in Figure 1C, shows total antibody response for all 4 strains, and are expressed as MN titers or fold change relative to the activity detected for each subject at baseline (D1). Significant baseline reactivity was detected with a median of 718.0 (range: 213.1 to 1771.0). MN titers of 7 out of 10 (70%) subjects increased on D15; median response was increased to 976.6 (range: 233.1 to 2146.0, *p* = 0.030) and fold change was 1.2 (range: 0.8 to 5.1, *p* = 0.027). On D91, responses in 7 out of 10 (70%) subjects were still elevated (median 919.4, range: 216.6 to 2146.0), with a fold change of 1.2 (range: 0.8 to 4.4), although the difference was no longer significant compared to D1. The individual trends of changes in MN antibody titers were shown in Appendix A.

Parallel studies evaluated serological responses by the Hemagglutination inhibition (HAI) assay which assesses the ability of test sera to prevent the agglutination of red blood cells [26]. As expected, the MN and HAI titers were highly correlated (*p* < 0.0001 and *r* = 0.7856) (Appendix A), and HAI total antibody response for all 4 strains followed a similar trend to MN, as shown in Figure 1D. The median baseline reactivity was 105.3 (35.3 to 273.1). One subject was excluded due to missing values, HAI titers of 7 out of 9 (77.8%) subjects were increased on D15 with an increased median reactivity 250.0 (range: 10.0 to 678.3) and a fold change of 1.3 (range: 0.1 to 6.4), although not statistically significant at this timepoint. On D91, all 10 (100%) subjects showed positive vaccine responses with further increased median response of 321.2 (range: 64.1 to 588.3, *p* = 0.002) and fold change of 1.9 (range: 1.1 to 6.4, *p* = 0.002). The individual trends of changes in HAI antibody titers were shown in Appendix A. When analyzing individual viral strains separately by both MN and HAI titers (Appendix A), the seroconversion rate for each strain (calculated as fold-change >4) ranged from 20% to 30%.

### 3.7. Balanced T Cell Reactivity at Baseline and Following Vaccination to the Four Different Strains

We examined the reactivity of the pre-existing cellular immune responses to the four different influenza strains at baseline. We found a fairly balanced level of pre-existing CD4 and CD8 baseline reactivity to all four viral strains (Figure 2A,B), each contributing to about 25% of the total number of responding T cells. This pre-existing reactivity is likely reflective of past exposure to related and cross-reactive influenza strains.

Intra-vaccine interference [27] is classically defined as a predominant response to one of the components in a multicomponent vaccine, leading to biased vaccine effectiveness towards one component rather than the multiple components as a whole. Here, we addressed this issue in the context of the multiple influenza strains included in Flucelvax vaccine. As shown in Figure 2C,D, the vaccine-specific (D15–D1) CD4 and CD8 reactivities remained fairly balanced, with significant and roughly equal contributions of the reactivity directed against each individual component. Although it appears to be more CD8 T cell responses to B/Iowa strain, however those were not statistically significant.

Next, we analyzed the reactivity following vaccination to each of the four strains. As expected, the CD4 and CD8 cellular responses showed similar trends including elevated reactivity on after 14 days and 90 days post vaccination to all 4 influenza strains (not all were statistically significant due to small sample sizes), with individual trends of responses for each strain summarized in Appendix A. Both the MN and HAI titers to individual viral strain also followed similar trends, as summarized in Appendix A. In conclusion, vaccine interference was not observed with Flucelvax vaccination.

### 3.8. “Ceiling Effect” Observed in Both Cellular and Humoral Vaccine Responses

The antibody “ceiling effect” that limits the serologic vaccine responses from repeated vaccination or previous influenza infection has been reported multiple times [28,29], however limited attention was paid to the cellular vaccine responses. Here, we investigated the relationships between the baseline levels of cellular and humoral immune responses and their changes after vaccination. Interestingly, as shown in Appendix A that both cellular and humoral vaccine responses shown inverse correlations with baseline levels of pre-existing influenza immune responses. For this analysis, we included the fold of changes (FC) of vaccine responses at both D14 and D90 post vaccination, and considered each time point for each individual separately. Appendix A showed that both CD4 (*p* = 0.0188, *r* = −0.5200) and CD8 (*p* = 0.0403, *r* = −0.4304) vaccine responses (represented as FCs after vaccination) were inversely correlated with baseline CD4 or CD8 responses to influenza virus strains. In addition, the humoral vaccine responses measured by FCs of both MN (*p* = 0.0443, *r* = −0.4541) and HAI (*p* = 0.0398, *r* = −0.4751) titers were also inversely correlated with the baseline influenza antibody levels (Panel C and D). Those observations suggested that the magnitude of both cellular and humoral vaccine responses might be limited by the pre-existing immune responses at baseline.

### 3.9. Type-Specific Immunodominance Patterns of CD4 and CD8 T Cell Responses

The fact that the reactivity is similar to the different strains, regardless whether measured before or after vaccination, could simply reflect a high level of similarity between the different MPs for the different strains. To gain more insight into this possibility, we analyzed the number of common peptides with 100% sequence similarity between different strains in the MPs, and found that in both influenza A and B strains, the core viral proteins are more conserved compared to the surface HA/NA (Appendix A). As expected, the “PA/PB1” and “PB1/PB2” MPs for the viral polymerase proteins are more conserved (with 56–62% common peptides in influenza A strains and 78–81% in influenza B strains), followed by the “other” MP for other viral proteins (NP/M1/M2/NS1/NEP) (with 27% common peptides in influenza A strains and 57% in influenza B strains), while the HA/NA pools were the least conserved (with 0% common peptides in influenza A strains and 22% in influenza B strains) among the different strains considered.

If the similar pre-existing reactivity was to be ascribed to sequence similarity between different viral strains, it would be expected to be highest in the more conserved PA/PB1, PA/PB2 and “other” pools, and the pattern of relative immunodominance, defined by which pool is dominantly recognized, to be similar across strains. The data presented in Figure 3A, show that this is not the case. Rather, the baseline reactivity detected in the two influenza A strains was mostly directed to the “other” MP (*p* < 0.0001), while in the two influenza B strains was mostly directed to the HA/NA MP (*p* < 0.0001). These data suggest that the similar baseline reactivity detected against the MPs corresponding to the four different influenza strains is not based on overall sequence similarity, but rather associated to a different immunodominance pattern for the A versus B strains.

In the case of CD8 responses, baseline reactivity was fairly evenly distributed across the various pools (Figure 3C), with the exception of A/H1N1, where the other protein (NP/M1/M2/NS1/NEP) pool accounted for a relatively higher reactivity than the other MPs. In summary, these data demonstrate a different pattern of immunodominance at baseline when influenza A and B strains are compared, especially with regards to CD4 reactivity.

### 3.10. Patterns of Reactivity of CD4 and CD8 T Cell to Influenza A and B Viruses Following Vaccination

Interestingly, type-specific immunodominance patterns of CD4 T cell responses remained unchanged after vaccination (Figure 3B), and the vaccine-specific CD4 T cells targeted predominantly the “other” MP of the two influenza A strains, and the HA/NA MP of the two influenza B strains. Again, the vaccine-specific CD8 T cell responses showed a pattern similar to the pre-existing responses at baseline (Figure 3D), predominantly targeting the internal viral proteins across all four strains of influenza A and B viruses, and resulting in significant and balanced total responses against both types of influenza viruses.

### 3.11. Differential Polarization of CD4 and CD8 Reactivity

In the next series of experiments, we evaluated the characteristics of the CD4 and CD8 T cell responses to the various components of the vaccine, with regards to the fraction of each specific cytokine or effector response. As shown in Figure 4A in the left panel, the baseline CD4 T cell responses before vaccination were strongly polarized towards IFNγ production indicating a Th1 response (*p* < 0.0001). This remained unchanged following vaccination (Figure 4C, left panel) with a similar pattern on D15 (*p* < 0.0001) and was seen across all four strains (Figure 4A,C, right panels) before and after vaccination.

In the case of influenza reactive CD8 T cells, both Granzyme B (*p* = 0.0002) and IFNγ (*p* = 0.0059) producing CD8 T cells dominated the pre-existing CD8 T cell responses at baseline (Figure 4B, left panel). This pattern remained unchanged following vaccination (Figure 4D, left panel) on D15 with predominant IFNγ (*p* = 0.0169) cytokine responses and elevated Granzyme B (*p* < 0.0001) activity. Similar patterns were observed across the four different strains (Figure 4B,D, right panels) before and after vaccination.

A similar analysis was performed in the case of the PT and CMV responses. In the case of CMV, IFNγ polarization was detected and remained unchanged before and after Flucelvax vaccination (responses for the four seropositive donors are shown in Appendix A. In the case of PT, it is known that individuals originally primed with the whole cell vaccine (wP) maintain a lifelong Th1 polarization, while individuals originally primed with the acellular vaccine (aP) maintain a lifelong Th2 polarization [30]. Indeed, in our case we observed a Th1 polarization in subjects that were (inferred by date of birth) likely primed with the wP vaccine, and a Th2 polarization in subjects that were (inferred by date of birth) likely primed with the aP vaccine. The polarization pattern was maintained in both cases before and after vaccination (Appendix A. In conclusion, the data presented here suggest that the T cell reactivity before and after vaccination follows the expected patterns of functional activity without affecting bystander cytokine responses to other microbial antigens such as CMV and PT.

### 3.12. MN and HAI Titers Correlate with CD4 T Cell Reactivity

As a further validation of the biological relevance of the analysis, we examined the association of MN and HAI titers with CD4 T cell reactivity and found significant correlation of both MN (*p* = 0.0186, *r* = 0.4270) and HAI (*p* = 0.0202, *r* = 0.4291) (Figure 5A,B), considering each visit of every individual as a separate data point. In addition, for both MN (*p* = 0.0147, *r* = 0.4412) and HAI (*p* = 0.0055, *r* = 0.5020) titers, the correlation was strongest with HA/NA specific CD4 responses as shown in Figure 5 (Panels C and D). Correlations were observed irrespective of the individual strains considered (not shown).

Further analysis showed (Figure 5E,F) that neutralizing titers of type B strains highly correlated with CD4 T cell responses from type B strains (*p* = 0.0030, *r* = 0.5235), while for the type A strains did not (*p* = 0.1269, *r* = 0.2850). This might indicate that, as opposed to type A, both type B specific CD4 T cell responses and neutralizing antibody responses mainly target the same HA-NA surface antigens, as shown in Figure 3 (panels A and B).

## 4. Discussion

The current study presents a comprehensive analysis of the pre-existing influenza-specific T cell reactivity and Flucelvax vaccine-induced immune responses. Our approach is comprehensive, as we evaluated both CD4 and CD8 responses, as well as serological responses to each of four strains contained in the vaccine. In addition, we report responses to minor, non-HA influenza proteins. The results demonstrated overall increased cellular and humoral immune responses on day 15 post vaccination which were still present on day 91 to each strain of both influenza A and B viruses.

While the efficacy of Flucelvax was reported to be 83.8% to vaccine-like strains [31], the seroconversion rate (fold-change > 4) only ranges from 36.6% to 49.2% for an individual viral strain [32]. Similarly, we found a seroconversion rate of 20% to 30% for each individual strain, implying a limited increase in antibody titers in this particularly study cohort likely due to high baseline responses; importantly, six individuals had received influenza vaccine other than Flucelvax in the previous season and a 4 months post-immunization period was required before enrollment in the study and vaccination with Flucelvax. As the “Ceiling effect” observed in our study, the baseline immune response levels limit both cellular and humoral vaccine responses. Future studies will be needed, particularly focused on vaccine constructs that differ substantially from previous seasonal vaccines.

Although subunit vaccines are generally considered to have limited capacity to boost cellular responses [23], recent studies have reported the induction of significant HA–specific CD4 T cell responses by Flucelvax [7,10]. While little is still known about the CD8 responses and their protein targets contained in this vaccine, here, we evaluated both CD4 and CD8 responses. While pre-existing CD4 responses were 4–5 times higher than CD8 responses in terms of magnitude, Flucelvax increased both overall CD4 and CD8 T cell reactivity, and both responses were consistently observed on day 15 and 91 post vaccination.

Vaccine interference could result from many factors [27], such as the nature and antigen content of the valences (e.g., recombinant proteins, semi-purified bacterial toxoids, inactivated purified whole viruses, live-attenuated viral strains), the type of additives (e.g., adjuvant, preservative, stabilizer), mode of use (e.g., immunization schedule and techniques), and vaccinee-related factors (e.g., pre-existing immunity, immune responsiveness and past microbiological history). Many of those factors have been reported to influence influenza vaccine effectiveness (VE), especially the intra-vaccine interference due to viral competition within the multivalent live-attenuated influenza vaccine (LAIV) leading to a lower VE against influenza A/H1N1pdm09 [33]. As a result, CDC decided not to recommend LAIV for use during 2016–2017 flu season [34]. Here, we found balanced immunogenicity of the Flucelvax vaccine, as both the pre-existing and vaccine-induced CD4 and CD8 reactivities were roughly equal to all four strains included in the vaccine, with no evidence of intra-vaccine interference.

Next, we evaluated the cytokine polarization of the cellular responses of Flucelvax, and consistent with what was found in pre-existing influenza-specific responses at baseline, vaccine-specific CD4 T cells produced primarily IFNγ cytokines and exhibited a dominant Th1 polarized profile. Vaccine-specific CD8 T cells produced predominantly IFNγ cytokines and displayed high Granzyme B activity, a phenotypic profile that was reported to be protective from not only symptomatic influenza infection, but also from severe complications in the 2009 H1N1 pandemic [35]. Although IFNγ-producing T cells have been studied in response to live attenuated and trivalent inactivated influenza vaccines [36], to our knowledge, the full polarization pattern following Flucelvax immunization has not been fully investigated prior to our study. Our results suggest that Fulcelvax vaccination induces a strong Th1 cell-mediated response, with limited IL-4 production, which together with Granzyme B activity, is considered essential for optimal protective immunity [37]. Currently, Flucelvax does not include any adjuvant [6]. To further improve its efficacy, future vaccine strategies need to consider not only the virus components, but also the adjuvant effect for driving the proper immune responses (e.g., alum is known to provoke strong Th2 responses, which is linked with vaccine enhanced disease in some cases) [38]. The functional assay used here could be applied to measure both the type and size of vaccine polarization if different adjuvants are introduced in future.

We further examined pre-existing and vaccine-induced responses to both HA and other influenza viral proteins (NA, M1, M2, NP, NEP, NS1, PA, PB1, and PB2) that may potentially elicit T cell responses [12,13]. Previous studies found that the majority (>80%) of pre-existing CD4 and CD8 T cell responses to influenza A virus strains (H3N2 and H1N1) are directed to the more conserved core proteins (NP/M/NS/PA/PB) and may mediate cross-reactive protection against different influenza A strains [39]. Our results found a similar immunodominant pattern in influenza A strains (H3N2 and H1N1), including CD4 and CD8 reactivity mostly directed to non-HA proteins, both at baseline and post vaccination. A different pattern was demonstrated to influenza B viral proteins, where there was a dominant HA/NA-related CD4-dependent humoral response. The existing reactivity for all the 4 strains observed at baseline implies either cross-reactivity with other strains (as mentioned above) or global widespread circulation of the strains contained in Flucelvax with previous natural exposure or previous vaccination [40]. In summary, cellular immune responses to influenza B viral proteins are largely understudied [41], and our results suggest that future vaccine studies should focus not only on the dominant responses to influenza A but also on the dominant responses to influenza B viruses.

This study was designed to characterize pre-existing and vaccine-specific immune responses rather than vaccine efficacy, and, therefore, there are limitations related to additional parameters to be correlated with the measure of vaccine-induced responses. First, the small number of study subjects did not allow to investigate the impact of age, gender, and ethnicity on influenza immunity, or the impact of vaccinations in previous years on the results obtained. Second, we designed peptides pools overlapping by 10 to ensure that all the possible epitope sequences recognized by CD4 and CD8+ T cells are captured, this did not allow us to compare to the peptides that share common sequences. Additionally, the cell number limitation did not allow to test for individual antigens or epitopes. Third, the current study focused on the immunogenicity and targets of the adaptive cellular and humoral immune responses of peripheral blood. In-depth proteomic, transcriptomic and functional analysis from blood, as well as other lymphoid tissues, will be addressed in consorted efforts by follow up studies.

## 5. Conclusions

Our extensive analysis of vaccine-specific immunity suggests that Flucelvax induces both humoral and cellular immunity, as well as a balanced multi-strain response in the absence of vaccine interference. Remarkably, the vaccine boosts responses to several different viral proteins. Induction of T cell responses in concert with serological response might be a factor in reducing severity of influenza-related complications [35]. The results of our study illustrate the benefit of a comprehensive functional evaluation of influenza vaccine responses, demonstrating the role of additional influenza proteins in boosting broad and balanced cellular immune responses that are desirable for protection from different influenza virus strains, and informing the development efforts for both seasonal and universal influenza vaccines.

## Figures and Tables

**Figure 1 vaccines-09-00426-f001:**
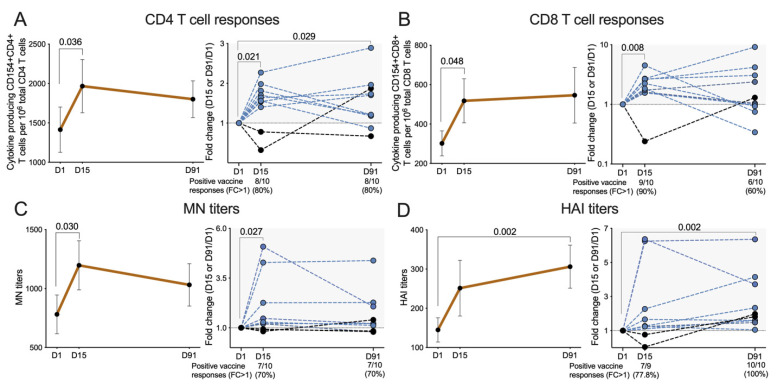
Cellular and humoral immune responses before and after vaccination. Cellular and humoral immune responses were measured at baseline (D1, pre-immunization), D15 (14 days post vaccination), and D91 (90 days post vaccination). (**A**,**B**) CD4 and CD8 T cell responses were represented by number of influenza virus reactive cytokine producing CD154+ CD4 or CD8 T cells per million of total CD4 or CD8 T cells. Left panels show the general trends of changes in CD4 and CD8 T cell responses for each visit (*n* = 10). Right panels show the vaccine induced changes for each individual person as represented by fold of change (FC) in CD4 or CD8 responses at D15 and D91 compared to baseline at D1 (FC = D15 or D91/D1). Percentages of positive vaccine responders (defined by FC > 1) at D15 and D91 were illustrated below the plots. (**C**,**D**) Humoral immune responses were measured by both microneutralization (MN) assay and hemagglutination inhibition (HAI) assay, and total responses of all 4 strains represented. Left panels show the general trends of changes in MN or HAI antibody titers for each visit (*n* = 10). Right panels show the vaccine induced changes for each individual person as represented by fold of change (FC) in antibody titers at D15 and D91 compared to baseline at D1 (FC = D15 or D91/D1). Percentages of positive vaccine responders (defined by FC > 1) at D15 and D91 were illustrated below the plots. (**A**–**D**) Normality of data distribution was accessed by Shapiro–Wilks test. Parametric data at D15 or D91 were compared to baseline (D1) by paired Student’s *t* test (two-tailed). Non-parametric data at D15 or D91 were compared to baseline (D1) by Wilcoxon test (two-tailed). Data were plotted as median with interquartile range for non-parametric data and mean with SEM for parametric data. *p* values < 0.05 were considered statistically significant.

**Figure 2 vaccines-09-00426-f002:**
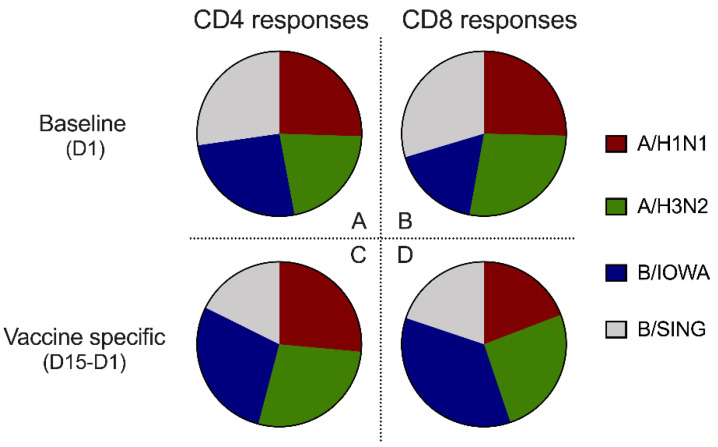
Contributions of all four influenza viruses strains to the cellular immune responses. (**A**,**B**) The relative contributions of all four influenza viruses strains to CD4 and CD8 responses at baseline (D1) for all 10 participants. (**C**,**D**) The relative contributions of all four influenza virus strains to vaccine-specific CD4 and CD8 responses (D15–D1) for all 10 participants. (**A**–**D**) CD4 and CD8 T cell responses were measured by number of influenza virus reactive, cytokine producing CD154+ CD4 or CD8 T cells per million of total CD4 or CD8 T cells, and vaccine specific T cell responses (D15–D1) were calculated by the CD4/CD8 T cells responses at D15 minus baseline CD4/CD8 responses at D1. Data were compared with one-way ANOVA adjusted with Tukey’s test for multiple comparisons, no statistical significance was found between strains.

**Figure 3 vaccines-09-00426-f003:**
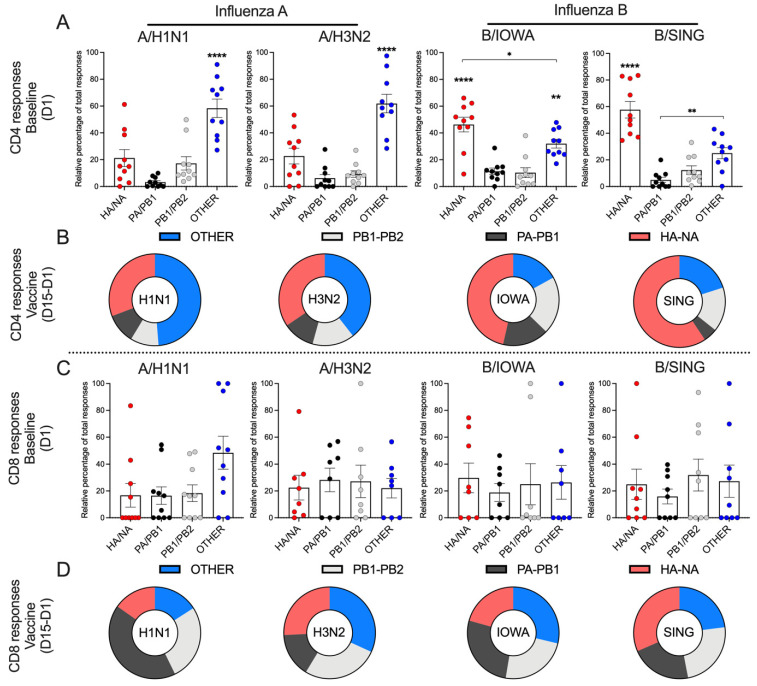
Protein immunodominance of CD4 and CD8 cellular immune responses. (**A**,**C**) The contributions of 4 major protein components (HA/NA, PA/PB1, PB1/PB2, OTHER–NP/M1/M2/NS1/NEP) of each influenza virus strain to the baseline (D1) CD4 and CD8 responses were summarized (*n* = 10). (**B**,**D**) The relative contributions of 4 major protein components (HA/NA, PA/PB1, PB1/PB2, OTHER–NP/M1/M2/NS1/NEP) of each influenza virus strain to the vaccine-specific (D15–D1) CD4 and CD8 responses were summarized (*n* = 10). Vaccine specific T cell responses (D15–D1) were calculated by the CD4/CD8 T cells responses at D15 minus baseline CD4/CD8 responses at D1. (**A**,**C**) Data were represented by mean ± SEM and were compared with one-way ANOVA adjusted with Tukey’s test for multiple comparisons. **** *p* < 0.0001, compared with all 3 other groups. ** *p* < 0.01, * *p* < 0.05.

**Figure 4 vaccines-09-00426-f004:**
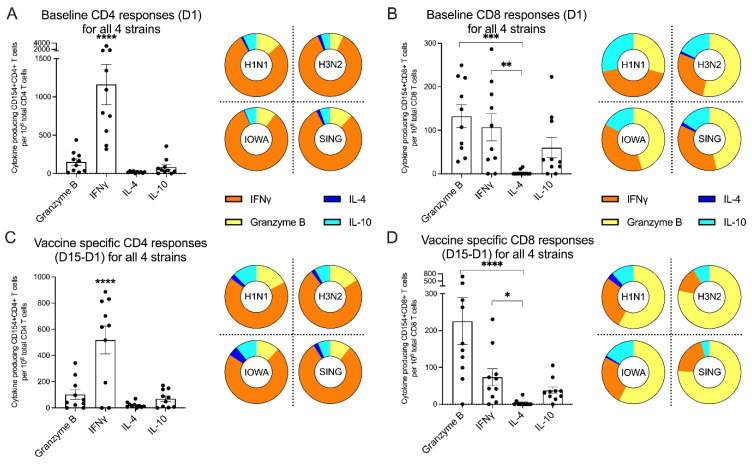
Effector-producing profile in CD4 and CD8 cellular immune responses. (**A**,**B**) The baseline (D1) CD4 and CD8 cytokine responses to all 4 strains of influenza viruses were summarized in the left panels (*n* = 10), and cytokine polarizations for each individual strain were represented in the right panels. (**C**,**D**) The vaccine-specific (D15–D1) CD4 and CD8 cytokine responses to all 4 strains of influenza viruses were summarized in the left panels (*n* = 10), and cytokine polarizations for each individual strain were represented in the right panels. (**A**–**D**, left panels) Data were plotted as median with interquartile range for non-parametric data and mean with SEM for parametric data. Normality of data distribution was accessed by Shapiro–Wilks test. Parametric data were compared with one-way ANOVA adjusted with Tukey’s test for multiple comparisons. Non-parametric data were compared with Kruskal–Wallis test adjusted with Dunn’s test for multiple comparisons. (**A**,**C**, left panels) **** *p* < 0.0001, compared with all 3 other groups. (**B**,**D**, left panels) **** *p* < 0.0001, *** *p* < 0.001, ** *p* < 0.01, * *p* < 0.05.

**Figure 5 vaccines-09-00426-f005:**
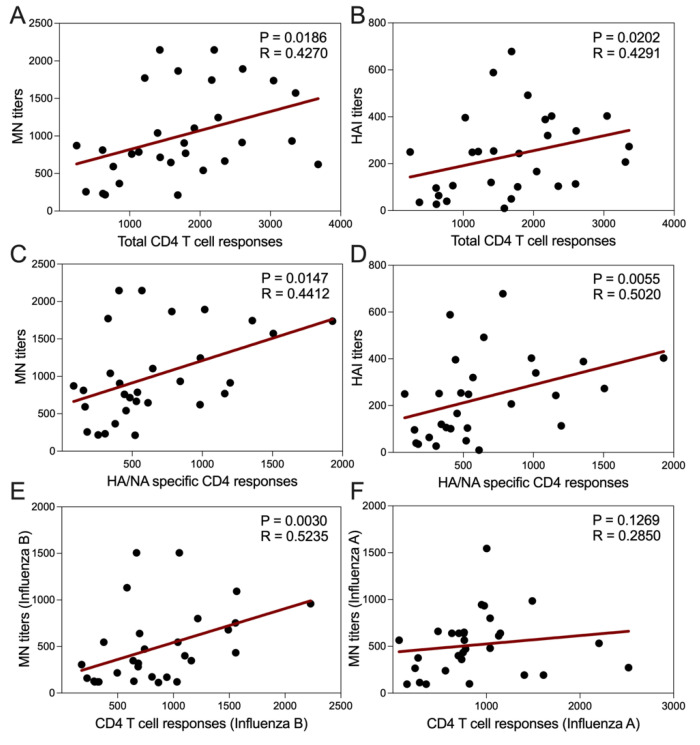
Correlation between cellular and humoral immune responses. (**A**,**B**) Total antibody titers for all 4 strains of influenza viruses measured by both microneutralization (MN) and hemagglutination inhibition (HAI) assays shown positive correlation with total CD4 T cell responses, which were measured by the number of influenza virus reactive cytokine producing CD154+ CD4 T cells per million of total CD4 T cells. (**C**,**D**) Total MN and HAI titers correlated with HA/NA specific CD4 responses better. (**E**,**F**) Influenza B specific MN titers correlated with Influenza B specific CD4 responses, however Influenza A specific MN titers did not correlate well with Influenza A specific CD4 responses. (**A**–**F**) Data from all 3 visits of 10 study subjects were included and correlation were calculated by Spearman correlation test. *p* values < 0.05 were considered statistically significant.

**Table 1 vaccines-09-00426-t001:** Demography of the cohort analyzed in this study.

Participant ID	Age at Immunization	BMI	Gender	Race
001	44	30.4	Male	Caucasian
002	47	24.5	Female	Caucasian
003	45	29.8	Male	Caucasian
004	23	25.8	Female	Caucasian
005	22	22.5	Male	Caucasian
006	33	22.1	Female	Caucasian
007	19	23.0	Male	Caucasian
008	33	21.6	Female	Caucasian
009	41	32.4	Female	African American
010	49	28.3	Female	Caucasian
Total	35.60 ± 3.54	26.04 ± 1.24	60% Female, 40% Male	90% Caucasian, 10% African American

BMI: body mass index, ID: identifier, average age and BMI are summarized as mean ± SEM (standard error of the mean).

## Data Availability

The datasets generated and analyzed in this study are available from the corresponding authors upon reasonable request. Likewise, biomaterials archived from this study may be shared for further research.

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
