# Peer review of "Balanced Cellular and Humoral Immune Responses Targeting Multiple Antigens in Adults Receiving a Quadrivalent Inactivated Influenza Vaccine"

_vaccines, 2021, doi:10.3390/vaccines9050426_

Round 1

Reviewer 1 Report

This manuscript by Yu et al describes the humoral and cellular responses to the Flucelvax quadrivalent vaccine for flu in a small cohort of 10 subjects. The authors find a balanced humoral response against all four influenza A and V viral strains, and a Th1 biased response with CD8 specificities largely to conserved core viral proteins. The data report on CD8 responses to core proteins have not previously been studied in this context.

The manuscript main body text and the results in figures and tables are clearly presented, support the conclusions made, providing new information about the Flucelvax vaccine with respect to cellular immunity with CD4+ HA/NA responses correlating to serum neutralization to influenza B strains, and revealing CD8+ specificities to core viral proteins. My only comments are therefore relatively minor in nature.

  1. Intro, 1st sentence. "...extremely simply structure" Besides the typo 'simply', can the authors be more specific or provide support for this claim. Flu virus structure would seem average in complexity to this reviewer, but it is too vague a statement so needs a more concrete description to make any definite claims.
  2. Intro. Ln 46-47 - authors mention antigenic drift and shift, should be supported with clear definition and reference, e.g. a recent review.
  3. Fig.1. The labels on the Y-axes is too small a font and should be enlarged.
  4. ln 313-315. "remarkable as subunit vaccines are generally not considered to be effective in inducing CD8 responses". However, isn't Flucelvax an inactivated vaccine, not subunit? If the CD8 response is so unusual the authors should do a more thorough job here and in the discussion with specific references to support their claim.
  5. ln 502-503. "...vaccination follows the expected pattern.. without bystander effects." This statement will likely be unclear for some readers. Can the authors be more explicit and describe the type of bystander effect that they are suggesting to have ruled out.
  6. ln 591-597. The authors point out adjuvant may play a role in Th1/Th2 polarization. The authors should state which adjuvant is used in Flucelvax, and cite information known about how it influences Th1/Th2 responses.
  7. ln 639-640. "...broad and balanced cellular immune responses... are essential for protection..." The authors should be more specific here. Isn't a narrow and somewhat skewed cellular immune response able to protect in some cases? Maybe the authors mean broad and balanced responses may be desirable...

Reviewer 2 Report

This report by Yu et al examines the humoral and CD4/CD8 T cell response following the quadrivalent, Flucelvax vaccine. They utilize overlapping peptide pools from proteins in each of the 4 vaccine strains to stimulate PBMC and measure cytokine production in samples before and after vaccination. They conclude that

  • T cells specific for all 4 strains were detected in the baseline samples.
  • CD4 T cells showed more reactivity to HA/NA of the influenza B strains, and other core proteins in influenza A strains.
  • There were more CD8 T cells specific for core proteins of both A and B strains relative to HA/NA.
  • There was an increase in CD4 and CD8 T cells following vaccination.
  • There was no evidence of vaccine interference between the different strains in the vaccine.

This is a comprehensive analysis of the T cell response to the different strains included in the vaccine. These findings are important as they provide insight into understanding the immune response to this vaccine. For the most part, the data presented support their conclusions. Questions and concerns are listed below:

  1. In figure 1, it would be helpful to see the lines of the individual responses in the left panel of each part to see if it is possible to assess whether the baseline levels impact the expansion. Also, it would be helpful to see how much it varies between individuals.
  2. For most of the analyses, it would be helpful to know if the baseline levels impact the amount of T cell expansion. In other words, do you observe similar T cell expansion of T cells that were more prevalent baseline as those that were less prevalent?
  3. Likewise, for the B cell response, is the sero-conversion so low because the levels of antibodies to these strains is already high?
  4. It appears that the T cell response to PT does increase after vaccination in about half of the individuals.
  5. In Figures 2, 3B and 3D, it is not clear what the D15-D1 represents. Is that the difference in the proportion of antigen-specific T cells at day 15 compared to day 1?  Or is it indicating the response after 14 days?  This is a major issue because it would alter the way the data are interpreted. I am assuming that the D1 is the baseline, and that the D15-D1 is the day 14 time point and not the fold-change between d1 and d15.  If, it is representing the fold change, then they need to show each time point individually.
  6. In figure 2, they conclude that the response to each virus is similar, but there appears to be more CD8 T cells specific for B/Iowa relative to the others.
  7. In Figure 3, it would be helpful to compare both the before and after infection data as scatter plots, rather than the before as scatter and the after as a pie chart.
  8. In Figure 3C and 3D, it appears that the H1N1 PA/PB1 cells are expanding after vaccination. Maybe the fold-change of T cells specific for each peptide pool could be presented.
  9. It would have been nice to have peptide pools without overlapping sequences to compare to the peptides that share common sequences.
  10. Do the CD4 T cell responses to the individual influenza A strains correlate MN titers or HAI for the individual strains (rather than grouping the response to the H1N1 and H3N2 together)?
  11. The authors conclude that CD4 reactivity to HA/NA correlated with antibody titers, but this does not appear to be true for the influenza A strains.

Reviewer 3 Report

The manuscript by Yu et al entitled, “Balanced cellular and humoral immune responses targeting multiple antigens in adults receiving a quadrivalent inactivated influenza vaccine”, sought to characterize and understand T cell responses following vaccination with a quadrivalent inactivated vaccine against influenza. Overall, the investigators found that after vaccination, there is an increase in neutralizing antibodies and an increase in the proportion of activated, influenza-specific, cytokine-producing CD4+ and CD8+ T cells. Some correlations were found between CD4+ T cell responses and neutralizing antibody levels. Investigators further showed that the T cell responses are generally even spread across the four strains in the vaccine and are primarily of the Th1-type at baseline and after vaccination. Moreover, T cell responses against influenza A were predominantly directed towards NP/M1/M2/NS1/NEP peptides while those against influenza B were directed towards HA/NA peptides. Overall, the investigators showed that there is an increased in humoral and T cell responses after vaccination and the quality of T cell responses appears to remain the same relative to baseline.

Major comments:

  • This manuscript is primarily descriptive and provides important information regarding T cell responses after responses. I personally feel that T cell responses after vaccination or virus infection are understudied but are valuable to understand and improve vaccination strategies.
  • I appreciated the use of CMV and Bordetella as control for non-specific immune activation since this phenomenon has been shown for other vaccines (e.g. BCG).

Minor comments:

Line 38: “simply” to “simple”

Line 167: “106” to “106

Line 185-190: These lines are more appropriate to be placed in the Results section.

Line 275-282: Please consider adding that it is interesting to note that one person did not have an elevated CD4 T cell reactivity at D15 or D91 timepoints.

Line 275-315: Is one of the two people who did not have increased CD4+ T cell reactivity at D15 and D91 the same person who did not have elevated CD8+ T cell responses on D15?

Line 383: Add comma to “influenza virus reactive, cytokine-producing”

Line 358-362: Please consider writing out some of the phrases in parentheses into a separate sentence to improve readability.  (E.g., These can be separated into other sentences: One subject was excluded due to missing values. HAI was not statistically significant at D15). You can apply this to the rest of the manuscript as well.

Overall:

- Please consider using one convention to specify timepoints to improve conciseness and readability. E.g., D1, D15, D91 or 0 days-, 14-days, and 90-days postvaccination instead of using both or either one interchangeably.

- Please review the numbering of your Supplementary Tables

- For all images: Please use the same ranges for the y-axis, especially when the measurement/readout is the same between plots. This makes it easy to compare magnitudes between the plots.

Reviewer 4 Report

In this manuscript, the authors introduced the humoral and cellular immune responses to Flucelvax. They concluded that the vaccine boosts responses to several different viral proteins. Induction of T cell responses in concert with serological response might be a factor in reducing severity of influenza-related complications. To my impression, the manuscript is presented in a well-organized and logical manner. All the experimental results obtained from their studies show reasonable consistency. In addition, these studies provide insightful knowledge of influenza vaccine responses and will contribute to further studies on its applications. I would therefore strongly recommend this manuscript for publication in Vaccines.
